**www.cambridge.org/qrd**

Computational modelling; target validation; drug site mapping

**Corresponding author:**
Agnieszka K. Bronowska;
Email: agnieszka.bronowska@ncl.ac.uk

S.X., L.F., P.Z., R.Z., K.S.H., J.V.d.S. and A.K.B. contributed equally to this work.

# 'Druggability' of the Per–Arnt–Sim (PAS) domains of human PAS domain kinase, a therapeutic target for metabolic and liver disorders

Shangze Xu[1,2], Lanyu Fan[1,3], Piotr Zaborniak[1], Ruidi Zhu[1], Haoyuan Ji[1], Kate S. Harris[1,4] (iD), João V. de Souza[1] and Agnieszka K. Bronowska[1,2] (iD)

[1]Chemistry—School of Natural and Environmental Sciences, Newcastle University, Newcastle, UK; [2]Newcastle University Centre for Cancer, Newcastle University, Newcastle, UK; [3]School of Engineering, Newcastle University, Newcastle, UK and [4]Translational and Clinical Research Institute, Newcastle University, Newcastle, UK

## Abstract

Per–Arnt–Sim (PAS) domain kinase (PASK) is a conserved metabolic sensor that modulates the activation of critical proteins involved in liver metabolism and fitness. However, despite its key role in mastering the metabolic regulation, the molecular mechanism of PASK's activity is ongoing research, and structural information of this important protein is scarce. To investigate this, we integrated structural bioinformatics with state-of-the-art modeling and molecular simulation techniques. Our goals were to address (1) how many regulatory PAS domains PASK is likely to have, (2) how those domains modulate the kinase activity, and (3) how those interactions could be controlled by small molecules. Our results indicated the existence of three N-terminal PAS domains. Solvent mapping and fragment docking identified a consensus set of 'druggable hot spots' within all domains, as well as at domain–domain interfaces. Those 'hot spots' could be modulated with chemically diverse small molecular probes, which may serve as a starting point for rationally designed therapeutics modulating these specific sites. Our results identified a plausible mechanism of autoinhibition of kinase activity, suggesting that all three putative PAS domains may be required. Future work will focus on validation of the predicted PASK models and development of small-molecule inhibitors of PASK by targeting its 'druggable hot spots'.

## Introduction

The protein kinase with Per–Arnt–Sim (PAS) domains (PASK) is a highly conserved serine/threonine kinase containing N-terminal PAS domains and a C-terminal kinase domain. It acts as a nutrient and energy sensor (Hurtado-Carneiro *et al.,* 2014, 2020) and can also respond to intracellular redox state, light, and oxygen (Dongil *et al.,* 2018; Pérez-García *et al.,* 2021; Hurtado-Carneiro *et al.,* 2021). In mammals, PASK senses nutritional status and contributes to the regulation of glucose homeostasis, energy metabolism, and oxidative stress (Hurtado-Carneiro *et al.,* 2021). PASK regulates glucagon and insulin secretion and is also a critical signaling regulator of the AMPK and mTOR pathways in the liver (Zhang *et al.,* 2015). In addition, PASK involvement in protein translation, cell differentiation processes, and epigenetic regulation has recently been described (Kikani *et al.,* 2019). Collectively, these make PASK an attractive therapeutic target, which has been recently validated for conditions such as non-alcoholic steatohepatitis and liver fibrosis (Zhang *et al.,* 2015; Swiatek *et al.,* 2020).

Despite its evolutionary conservation and established role in coordination between cellular metabolism, oxidative stress, and metabolic demand, the molecular mechanism underlying PASK regulation remains ongoing research (Hurtado-Carneiro *et al.,* 2020). The crystal structure of the kinase domain of human PASK shows that this domain adopts an active conformation and has catalytic activity in the absence of activation loop phosphorylation (Kikani *et al.,* 2010). It has been therefore suggested that the kinase domain of PASK is autoinhibited by one of its N-terminal PAS domains. The current understanding of PASK activation is that under standard conditions, PASK remains autoinhibited, and the interaction between certain small molecules (such as nutrients or metabolites) and the PAS domain terminates the PASK's autoinhibited state by unblocking the kinase domain (Figure 1). This may lead to subsequent phosphorylation of PASK's substrates and regulate their downstream effectors (Hurtado-Carneiro *et al.,* 2021).

Despite ongoing efforts, the structural biology and precise mechanism of PASK regulation remain poorly understood. Likewise, the endogenous small molecules modulating PASK are still unknown, and the relationship between PASK and oxidative stress requires better understanding. Since PAS domains are structurally conserved, versatile sensor domains reported to detect

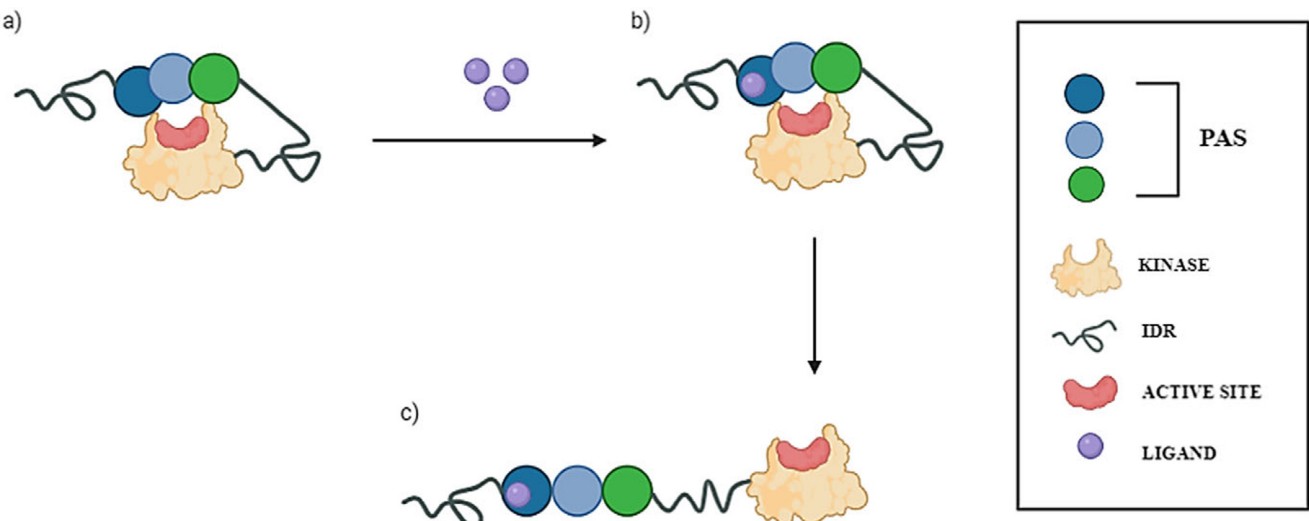

**Figure 1.** The current proposed model of PASK activation. Putative PAS domains of human PASK kinase are presented as circles (blue, aqua, and green). The kinase domain is presented as a yellow oval, and the active site where ATP is bound is depicted as a red circle. Intrinsically disordered regions (IDRs) are depicted as springs. (*a*) Under resting conditions, PASK assumes a closed, autoinhibited state, wherein PAS domains block the active site of the kinase domain. (*b*) In the presence of small-molecule ligands (e.g., nutrients, metabolites) depicted as purple circles, one of the PAS domains binds to a ligand, causing conformational changes within the domain and affecting its interactions with the kinase domain. (*c*) This conformational change leads to unblocking of the kinase domain, thus rendering PASK active.

hypoxia, redox state, and a broad spectrum of chemically diverse small molecules (e.g., metabolites, pollutants, and drugs) (Poch *et al.,* 2019; Arnaiz *et al.,* 2021; Mullard, 2021), it may be speculated that PAS domains of PASK are sensor domains, crucial for the regulation of this kinase.

PAS domains are small (~100 residues) motifs, present in all kingdoms of life (de Souza *et al.,* 2019). In eukaryotes, PAS domains are found in several classes of signaling proteins, including transcriptional regulators, transmembrane channels, and kinases (de Souza *et al.,* 2019). Despite their primary sequence variability, these domains have a highly conserved three-dimensional structure, with a hydrophobic cavity where small-molecule ligands may bind. In other mammalian PAS domain sensors, such as the aryl hydrocarbon receptor (AhR), such a binding event initiates a conformational change that spreads over all the protein and affects protein regions distal to the binding event. This may have varied results, such as dissociation of the PAS domain from its cognate binding partner (e.g., Hsp90–AhR complex), formation of a new complex (e.g., AhR–ARNT), and activation (e.g., transcription regulator function of AhR, NPAS, and HIF proteins) (Greb-Markiewicz and Kolonko, 2019; Soshilov *et al.,* 2020; Janssens and Stove, 2021).

Amezcua *et al.* (2002) studied the conformational changes within the first PAS domain of human PASK by combining structural biology, biophysical, and biochemical techniques. By solving an NMR structure of this domain (PDB code: 1LL8, residues 131–237), they showed that the domain contains two unusually flexible segments. The locations of those segments indicated that these segments might serve as functionally relevant binding interfaces. NMR-based screening of a library containing over 750 small molecules showed that this domain binds a diverse range of small molecules within its binding site. Furthermore, their work showed that the kinase domain of PASK interacted with the first PAS domain, providing the initial direct biophysical observation of protein–protein interactions (PPIs) within PASK. They concluded that the PAS and kinase domains of PASK are functionally and structurally linked, suggesting a regulatory pathway for small organic molecules to modulate the enzymatic activity of PASK. This study provided a robust proof-of-concept for PASK allosteric regulation occurring via PAS domains. However, the N-terminus of PASK is very long (~950 residues), where several PAS domains have been mapped by bioinformatic approaches (Figure 2). Even though the NMR structure of the first PAS domain of human PASK has been solved (Amezcua *et al.,* 2002), the remainder of the N-terminal region of PASK, including the number, location, and potential 'druggability' (that is ability to bind small molecules, such as drugs or metabolites) of those elusive PAS domains remains unexplored.

In this work, we have combined complementary computational approaches, to address the following questions: (i) how many PAS domains are likely to span the N-terminus of PASK kinase; (ii) what is their overall structure and how flexible are these domains; (iii) can all of these domains bind small-molecule ligands, and what is the chemical nature of those putative ligands; (iv) which of those PAS domains is the most likely to interact with the kinase domain, and could these putative interactions be modulated by small molecules?

First, using several orthogonal bioinformatics methods, we mapped the location of putative PAS domains. Three-dimensional models of those domains were calculated using a combination of homology modeling via Phyre2 (Kelley *et al.,* 2015), alongside modeling using AlphaFold (Jumper *et al.,* 2021). Next, we assessed the overall stability and quality of those models by all-atom molecular dynamics simulations in explicit water and excluded those models that showed abnormal energetics, had an unusual number of Ramachandran outliers, or unfolded during the simulations. The resulting model indicated an intrinsically disordered N-terminus, three consecutive PAS domains, and a long, partially disordered linker joining PAS domains and the kinase domain of PASK. The models of the first two domains (PASK1 and PASK2) are consistent with structural and published data (Amezcua *et al.,* 2002; Kikani *et al.,* 2019); however, details of the architecture of the third PAS domain require further investigation. Next, we assessed interactions between PAS and kinase domains by protein–protein

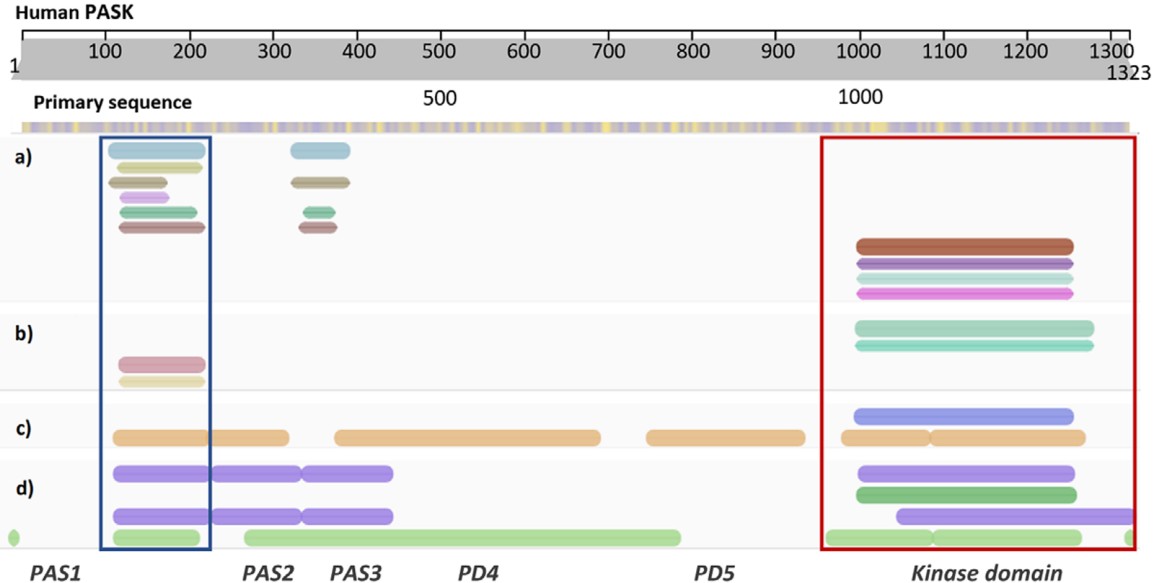

**Figure 2.** Diagram showing annotation of human PASK domains by different bioinformatics resources. (*a*) Consensus domain annotation of N-terminal PASK segment by established structural bioinformatics databases: InterPro – blue, TIGR – khaki, SMART – brown, Prosite – lilac, Pfam – bottle green, and CDD – ochre. Kinase domain has been annotated separately and the colors correspond to dark red – InterPro, purple – SMART, aqua – ProSite, and magenta – Pfam. (*b*) Consensus homologous superfamily: cyan and aqua – protein kinase-like (InterPro); red – PAS superfamily (InterPro) and orange – PYP-like sensor/PAS (InterPro). (*c*) Predicted 3D structural regions annotated by Genome3D [35]: blue – PHYRE 2 [36] model; brown – structural regions detected by DomSerf [37]. (*d*) Predicted domains annotated by Genome3D: purple (top) – FUGUE SCOP-based classification; dark green – kinase domain reported by PHYRE2; purple (bottom) – FUGUE CATH-based classification; light green – predicted by pDomTHREADER. Two domains with resolved experimental structures are marked by rectangular selection: PAS1 domain – blue, and protein kinase domain – red. Domains PAS2–PD5 are putative, deduced from the consensus between several predictive tools.

docking. These results suggest an explanation of how the kinase domain may be activated by small-molecule binding to N-terminal PAS domains. 'Druggability' mapping of individual PAS domains, free and bound to the kinase domain, indicated that these PAS domains may be functionally non-redundant and serve as sensors for chemically diverse ligands. Our results also indicate the presence of interfacial 'druggable' sites at autoinhibited PASK, which may pave the way for development of new drugs to inhibit PASK via an allosteric mechanism. Finally, our results suggest that metformin, a first-line drug to treat type 2 diabetes (Wang *et al.*, 2017), may inhibit PASK by acting as a 'molecular glue' between the PAS1–PAS2 segment and the kinase domain. This is likely to shed light on metformin's mechanism of action, which is not fully understood (Agius *et al.*, 2020).

## Materials and methods

### Structural bioinformatics

A collection of orthogonal tools was used to predict, cross-check, and annotate putative PAS domains within the N-terminal portion of human PASK (UniProt ID: Q96RG2). For consensus domain annotations, the following tools were used: Interpro (Blum *et al.*, 2021), TIGR (Haft *et al.*, 2001), SMART (Letunic *et al.*, 2021), Prosite (Sigrist *et al.*, 2013), Pfam (Mistry *et al.*, 2021), and the conserved domains database (CDD) (Lu *et al.*, 2020) In this work, all alignment parameters used their default parameters. For sequence/structural annotation, Genome3D was employed (Lewis *et al.*, 2015), using its default parameter. For prediction of putative domains and structured regions within PASK sequence, we used a combination of Phyre2 homology modeling webserver (Kelley *et al.*, 2015), DomSerf (Buchan *et al.*, 2013), FUGUE (Shi *et al.*, 2001), and pDomThreader/PSIPRED (Lobley *et al.*, 2009). All

algorithms for prediction of domains used their default parameters. The outputs were manually curated, analyzed, and integrated into a single comprehensible consensus output.

### Molecular modeling

All plausible domains were extracted from the full-length Alpha-Fold model (AFQ96RG2) generated from the structure available at Uniprot (Uniprot code Q96RG2) and subjected to 10,000 of molecular mechanical energy minimization using Amber99SB-ILDN force field (Lindorff-Larsen *et al.*, 2010) using GROMACS (van der Spoel *et al.*, 2005). Homology models were calculated using Phyre2 (Kelley *et al.*, 2015) and Swiss-Model (Waterhouse *et al.*, 2018) using the following templates: PDB code 1LL8 for human PAS1 (sequence similarity = 60%); PDB code 5SY5 (sequence similarity = 30%); for human PAS2; PDB code 4GW9 for human PAS3 (sequence similarity = 29%); and PDB code 6PH4 for PAS4 (sequence similarity = 31%). For the construct consisting of both PAS1 and PAS2, PDB code: 5SY5 was used as a template. These templates were selected using the following criteria: highest sequence similarity between the target and the template and the lowest number of outliers from the Ramachandran plot.

### All-atom molecular dynamics simulations

To address structural stability, energetics and intrinsic dynamics of each putative PAS domain within human PAS–kinase, all-atom molecular dynamics (MD) simulations were performed on the following distinct regions of PASK: PAS1 (residues 124–237), PAS2 (238–335), PAS1–PAS2 construct (residues 124–335), PAS3_homm (residues 337–437), PAS3_AF 3 (residues 343–400 and 885–982), and PD4 (residues 485–592).

All simulations were carried out in GROMACS 2016.1 (van der Spoel *et al.*, 2005), using the AMBER99SB-ILDN force field with the TIP3P water model (Lindorff-Larsen *et al.*, 2010). In each simulation, the protein under investigation was placed in a center of a cubic simulation box. The distance between the box edge and the protein was set to 1 nm. The box was filled with solvent (water) and counter-ions ($Na^+$ and $Cl^-$) were added to a concentration of 0.1 M to maintain charge neutrality of the simulation unit and to mimic their physiological concentration. Next, molecular mechanical energy minimization was performed first, for 1,000 cycles of steepest descent, and after that, the algorithm was switched to a conjugate gradient. The minimization step size was set to 0.01 nm, and the maximum number of steps performed was 50,000. The energy minimization stopped when the maximum force of the system was lower than 1,000 kJ $mol^{-1}$ $nm^{-1}$ using the Verlet cutoff scheme (Verlet, 1967). The particle mesh-Ewald (PME) was set to long-range electrostatic interactions (Darden *et al.*, 1993), and the short-range electrostatic and van der Waals cutoff was set to 1.0 nm.

Energy-minimized systems were equilibrated, including the following steps: (1) NVT equilibration for 20 ps, with 2 fs time step, wherein the system has been heated and the target temperature was set at 300 K, by using a modified Berendsen thermostat ($\tau = 0.1$ ps) (Berendsen *et al.*, 1984). The constraint algorithm used was LINCS (linear constraint solver) for all bonds in the protein (Hess *et al.*, 1997). The Verlet cutoff scheme was set to 1.0 nm for non-bonded short interaction, and long-range electrostatics were set to PME. (2) NPT equilibration (20 ps), wherein pressure was kept constant at 1 bar by Parrinello–Rahman isotropic coupling ($\tau = 2.0$ ps) to a pressure bath (Parrinello and Rahman, 1981). (1) and (2) included positional restraints on backbone atoms, to prevent unfolding and other possible artifacts of rapid heating and equilibration. (3) 60 ps of NPT equilibration, with no positional restraints. (4) The production MD simulations were performed for 500 ns in triplicates. The coordinates and energies were saved every 10 ps. Collectively, all production simulations amounted to 9 ms.

### Data analysis

Analysis of MD production trajectories was performed using GROMACS tools, FTMap webserver (http://ftmap.bu.edu/login.php) (Kozakov *et al.*, 2015), and MDAnalysis package (Michaud-Agrawal *et al.*, 2011; Gowers *et al.*, 2016). GROMACS tools included RMSD (root-mean-square deviation) calculations to assess the structural stability of the domain of interest, RMSF (root-mean-square fluctuation) calculations to assess the flexibility per-residue, PCA (principal component analysis) to assess low-amplitude correlated motions, and cluster analysis to evaluate heterogeneity of produced molecular ensembles. H-bond calculations was used as a metric of internal stability of the system, and its internal potential energy calculated via gmx energy. For each system, all trajectories were concatenated prior to the cluster analysis (Daura *et al.*, 1999) and PCA. The outcomes of PCA and cluster analysis, namely top-three clusters, and clusters representative to PC1-PC3, respectively, were subjected to FTMap: a computational mapping server that identifies binding hotspots in proteins (Kozakov *et al.*, 2015) and a widely used tool on which our group has used and tested (Sabanes *et al.*, 2019). Clustering ensemble similarity (CES), a convergence analysis tool from MDAnalysis package (Michaud-Agrawal *et al.*, 2011) was used for assessing the convergence of the systems. We monitored the cluster evolution through time using MDAnalysis alongside RMSD. The

tool splits the trajectory into different windows. the CES value was calculated each time a new conformation space is formed by the current window plus the previous windows, and it was compared with the conformation space formed by the entire trajectory. Faster drop of CES value to 0 means the earlier convergence of the trajectory. Results for CES analysis is shown in Supplementary Figure S14.

### Protein–protein docking

Protein–protein docking was performed with ClusPro (https://cluspro.bu.edu/). We used rigid docking, as described by Kozakov *et al.* (2017). Fast-Fourier transform (FFT) employed within ClusPro enables the sampling of billions of putative complex conformations (Desta *et al.*, 2020). This makes the algorithm highly suitable for global docking without any *a priori* information on the structure of the complex. The performance validation of ClusPro is showed in Supplementary Figure S13.

Docking of PAS and kinase domains of human PASK was carried out sequentially. First, we assigned the crystal structure of the kinase domain (PDB code: 3DLS) as the 'receptor' and all PAS domains as 'ligands' of binary global docking. We used the highest-populated cluster from MD simulation of each PAS domain as a representative 'ligand' frame. Two alternative models of PAS3 were considered. Flexible loops and annotated intrinsically disordered regions were excluded from the calculations in order to reduce noise within the generated data. Each docking run consisted of three following stages: (1) rigid body docking by sampling billions of conformations, applying no geometric restraints; (2) root-mean-square deviation (RMSD) based clustering of the 1,000 lowest energy structures generated to find the largest clusters that will represent the most likely models of the complex, and (3) refinement of selected structures using molecular mechanical energy minimization. The rigid body docking step used PIPER, a docking program based on the FFT correlation approach (Kozakov *et al.*, 2006). In each run, the 10 best-ranking models were visually inspected.

Furthermore, we used the same procedure to dock: PAS1–PAS2 segment to the kinase domain; PAS2 to the PAS1-kinase domain complexes, PAS1 to the PAS2-kinase domain complexes; PAS3 to the PAS1–PAS2-kinase ternary complexes; PAS2 to the PAS1–PAS3-kinase ternary complexes; and PAS1 to the PAS2–PAS3-kinase ternary complexes. To evaluate the propensity of PAS domains binding to each other, we also calculated PAS–PAS2–PAS3 complexes using the same procedure and subsequently used top-five complexes for the binary docking to the kinase domain. Each calculation involving PAS3 domain included two alternative models (AlphaFold and homology). In total, 24,000 complexes were calculated and evaluated. In each run, the 10 best-ranking models were visually inspected.

Next, we selected those models that satisfied the distance requirement of PAS1–PAS2 and PAS2–PAS3 being less than 1.2 nm apart. This is required, as the interdomain PAS1–PAS2 and PAS2–PAS3 linkers are very short. Such an approach yielded 19 models, with different relative orientation of the domains, and different numbers of PAS domains, interacting with the kinase domain simultaneously. Those complexes were energy-minimized, and the three best-ranking models were selected. Out of these, we prioritized the model that matched best the experimental data available (Amezcua *et al.*, 2002; Kikani *et al.*, 2019).

### Small-molecule docking

The MiniFrag library (O'Rreilly *et al.*, 2019) containing 80 chemically diverse, drug-like fragments in SDF format (e.g., phenol, 2-bromopyridine, piperazine, pyrrolidin-2-ylmethanol) was obtained from Enamine (www.enamine.net). The following compounds were added to the library: glucose, fructose, n-hexane,

cyclohexane, cyclohexanone, glycerol, glycerol-3-phosphate, arginine (side chain), glutamate (side chain), histidine (side chain), lysine (side chain), metformin, dimethyl fumarate, and monomethyl fumarate. The resulting set, denoted as AugMiniFrag, was used in small-molecule docking calculations.

For each PAS domain, mini-ensembles derived by clustering (PCA analysis and RMSD clustering, using 0.35 nm as the cutoff; six distinct conformations), were subjected to the molecular docking procedure using SeeSAR version 11.2 with HYDE scoring function (Reulecke *et al.,* 2008). To evaluate potential PPI sites at kinase–PAS interface, we used the best-scoring complex from protein–protein docking, after MM energy minimization. To map the consensus sites, we searched for the binding sites by FTMap and by the volume search tool in SeeSAR. Only those sites that were reported by both tools were considered for docking. For each consensus site, the AugMiniFrag set was docked using non-covalent docking, maximum 200 poses for each fragment, with high clash tolerance selected. Small molecules with calculated binding affinities worse than high μM range (>500 μM) and torsional strains were discarded. The remaining sets were used for mapping of the consensus 'hot spots' within each predicted PAS domain and the PAS–kinase complex.

## Results and discussion

### Annotation of putative PAS domains within human PASK

While the location and three-dimensional structure of the first PAS domain (PAS1; PDB code: 1LL8; residues 131–237) and the kinase domain (PDB code: 3DLS; residues 977–1300) of human PASK are confirmed (Amezcua *et al.,* 2002; Kikani *et al.,* 2010), the molecular architecture of the region linking those distal domains has been largely unexplored. Due to the length of this sequence, and the presence of intrinsically disordered regions (IDRs) within, the total number of predicted domains reported and their precise location vary depending on the bioinformatics tools employed. For example, domain annotation within InterPro reports only two PAS domains (spanning residues 121–232, corresponding to PAS1, and 335–402, corresponding to PAS3, Figure 2*a*), while SCOP and CATH-based classification predicts three PAS domains close together, including experimentally confirmed PAS1 (Figure 2*d*).

We evaluated those predictions by (i) generating 3D models of all predicted segments, and (ii) investigating the stability of those hypothetical domains by subjecting the calculated models to 500 ns all-atom MD simulations (Supplementary Figures S1–S8). Each run was repeated three times, and the structural features, intrinsic dynamics and stability overall were evaluated. For quality control purposes, we subjected the experimentally confirmed PAS1 domain to the same procedure (Supplementary Figure S2).

Prior to MD simulations, we compared our generated models with predictions using AlphaFold (Jumper *et al.,* 2021). The Alpha-Fold model of human PASK consists of three PAS domains and the kinase domain, with PAS1 and the kinase domain matching well with the experimental structures, and the second PAS2 domain matching the model calculated by homology modeling techniques. However, the third PAS domain predicted by AlphaFold consists of two non-consecutive sequence strands (Figure 3*a*): 354–397, which is part of PAS3 modeled by homology modeling and threading approaches used by Phyre2 (Figure 3*b*), and residues 885–932, corresponding to the PD5 region, shown in Figure 2. The confidence score assigned to this PAS3 model was low, particularly this non-consecutive region (70 > pLDDT >50). Nevertheless, we have

included this model in further analysis by all-atom MD simulations. In addition, AlphaFold did not detect the putative fourth PD4 domain (residues 485–592, Figure 3*c*) which has been modeled by homology modeling (Figure 3*d*). Again, we have analyzed this by all-atom MD. Attempts to model PD5 as a standalone domain by homology modeling and threading approaches failed to produce a good quality model.

### All-atom MD simulations indicate three PAS domains within human PASK

All initial models of individual domains were subjected to 500 ns unrestrained equilibrium MD simulations. Data analysis showed that, while PD1–PD3 were stable during the simulation time in all replica runs (Figure 3*e*), PD4 partially unfolded in two out of three replicas (Figure 3*g*). This is caused mainly by losing its internal polar contacts in favor to the solvent. Analysis of internal potential energies of all individual domains further suggests that PD4 is very unlikely to retain the PAS-like fold calculated by the homology modeling technique (Supplementary Figure S5). PAS3 domain modeled by AlphaFold remained highly stable (Figure 3*f*), not showing any major conformational changes during the simulation after achieving a stable configuration. PAS3 domain modeled by Phyre2 (homology modeling) remained highly stable in two out of three replicas (Figure 3*e* and Supplementary Figure S4), achieving a stable configuration during first 50 ns of the simulations. In one of the replicas, major conformational changes within the 'gate-keeper' loop occurred, yet without unfolding (Supplementary Figure S4).

To further investigate the exact location, composition, and stability of the PAS3 domain, we also simulated an AlphaFold model, as an alternative to the homology model. The comparison of these two models is shown in Figure 3*f*. Despite being composed of two non-consecutive sequence strands (Figure 3*a*) the model retained features of the canonical PAS domain fold, such as the L-loop with the 'gatekeeper' tyrosine residue (Y389), and the core comprised of five antiparallel beta-sheets. There was a significant difference between average values for their RMSD (0.2 nm for the AlphaFold model and 0.8 nm for the homology model) from the starting structure. Both models equilibrated fast and reached their stable conformations before 50 ns of MD simulation (Figure 3*f*). There were several highly fluctuating residues within the AlphaFold model (Supplementary Figure S10), but the fluctuations resulted from non-constrained terminal regions (residues L885–L982 and C343–L400), and those fluctuations are very likely introduced by the simulation setup. However, calculations of the overall number of hydrogen bonds showed that the AlphaFold model was considerably more stabilized by intramolecular H-bonds than the homology model. On average, the analysis showed 75 ± 4 H-bonds in the AlphaFold model, compared to 60 ± 6 H-bonds within the homology model of PAS3. The analysis of potential energies for both models favored the AlphaFold model: −5,000 ± 300 kJ/mol on average, compared with −3,000 ± 400 kJ/mol for the homology model. These results strongly indicate that the existence of a third PAS domain within PASK is very likely, since its prone to fold into a highly energetically favorable conformation with a high number of well-organized internal hydrogen bonds.

The MD simulations of both PAS3 models show that the low-confidence model calculated by AlphaFold, comprised of two non-consecutive sequence strands, is more plausible. Nevertheless, we subjected both PAS3 models to the 'druggability' assessment and subsequent further analysis.

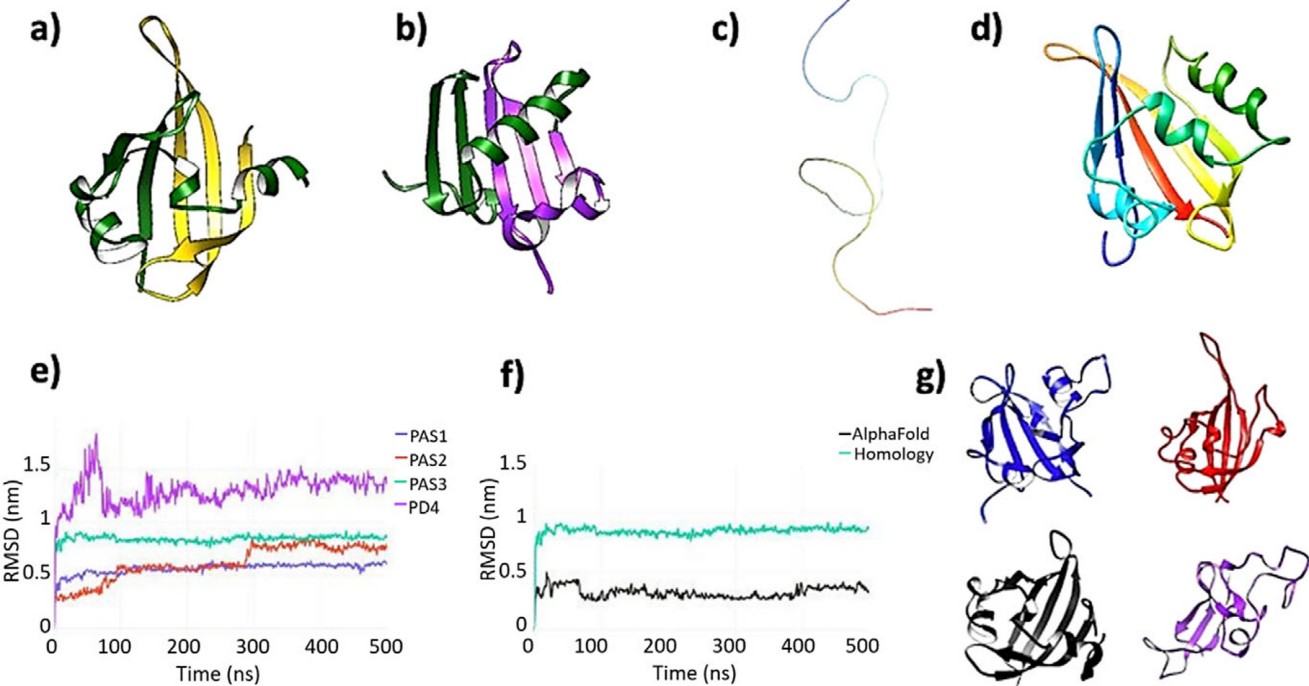

**Figure 3.** (*a*) AlphaFold model of the putative third PAS domain of human PASK. AlphaFold modeled PAS3 as two non-consecutive sequence strands folded in a single globular domain. The N-terminal strand, comprised of residues 343–393, is colored dark green, while the remaining part, comprised of residues 885–932, is colored gold. This latter segment corresponds to the putative domain PD5, highlighted in Figure 2. (*b*) Homology model of PAS3 domain. Both Swiss-Model and Phyre2 modeled PAS3 as folded from the consecutive sequence strand (residues 343–435, template: 4GW9:D). The RMSD value between AlphaFold PAS3 model and homology model of PAS3 is 14.393 angstroms (Supplementary Figure S15). To make this panel consistent with the AlphaFold model, residues 343–393 of the domain are colored green, while the remaining part (residues 394–435) are colored purple. (*c*) AlphaFold did not model a putative fourth PAS domain (PD4), representing this region as an unfolded, random coil. The strand (residues 485–592) is colored as a gradient from the N-terminus (blue) to the C-terminus (red). (*d*) Phyre2 modeled PAS4 as a folded, globular domain, resembling PAS-fold (template: 6PH4). The color scheme used in this panel is the same as in panel *c*. (*e*) Root-mean-square deviation (RMSD) plots of four putative PAS domains: PAS1 – blue, PAS2 – red, PAS3 – aqua, and PD4 – purple. (*f*) Comparison of RMSD calculations for two models of PAS3 domain: calculated by AlphaFold (black) and calculated by Phyre2 (aqua). (*g*) Representative frames extracted from four domains under investigation; the color scheme is consistent with panels *e* and *f*: PAS1 – blue; PAS2 – red; PAS3 (AlphaFold) – black, and PD4 – purple. PAS1–PAS3 remained stable and well-folded over the simulation time, while PD4 partially unfolded.

To assess the quality of our simulated ensembles and to evaluate the extent of sampling at 500 ns time scale, we compared the simulation results of the PAS1 domain with the ensemble derived from the solution NMR data available for this domain (Amezcua *et al.,* 2002). We found the overall trends observed in both ensembles to be consistent, with Pearson correlations between RMSF calculated from each three MD simulations replicas and NMR ensemble of 0.88 (replica 1), 0.71 (replica 2), and 0.77 (replica 3). Residues from helix F and loop FG (residues 175–200) were highly flexible in both ensembles (Supplementary Figure S2). Residues 185–194 and termini showed the highest root-mean-square fluctuations in our MD simulations at 500 ns time scale. Amezcua *et al.* (2002) showed that these residues exhibited elevated values of the spectral density function $J(\omega)$ at high frequencies [$J(0.87\omega H)$] and correspondingly decreased values at lower frequencies [$Jeff(0)$], indicating that these residues are flexible on fast (ps-to-ns) timescale. Residues 175–184 and 195–200, at both ends of the FG loop, showed average $J(0.87\omega H)$ and elevated $Jeff(0)$ values, which indicates that these regions undergo conformational changes on a slower (microsecond to millisecond) timescale. Interestingly, PCA analysis of our 500 ns simulations indicates slow, correlated motions within these regions of PAS1 (Supplementary Figure S3). Collectively, this is consistent with available NMR data and validates our methodology.

As the linker connecting PAS1 and PAS2 domains is very short (3–4 residues), a segment containing both domains and the linker was subjected to the same procedure (500 ns MD simulation, followed by PCA and cluster analysis). The results are consistent with the simulations of isolated PAS1 and PAS2 domains (Supplementary Figures S6–S8) and indicate the propensity of those two domains to change their orientation, but not to the 'collapse' of PAS1 on PAS2, which could be attributed to a very short linker (Supplementary Figure S9). Furthermore, the intrinsic flexibility of a linker allows domains to re-orientate themselves upon binding to the kinase domain (Figure 4).

Simulations of the PD4 region indicated that this region is not fully unfolded, as suggested by AlphaFold (Figure 3*c*), and retains a residual fold comprised of three beta-sheets (residues 552–590), resemblant of SH3-like domain structure. However, the biological significance of this putative domain is uncertain and requires further study, e.g., deletion of this region from the recombinant construct and assessment of the protein's function.

In summary, our approach indicates the presence of two consensus N-terminal PAS domains in addition to the PAS1 domain, which has been experimentally confirmed resolved by the solution NMR (Amezcua *et al.,* 2002). The third PAS domain is likely to be composed of two non-consecutive sequence segments, as suggested by AlphaFold. Nevertheless, an alternative, consecutive model, calculated by homology modeling techniques, seemed to be stable.

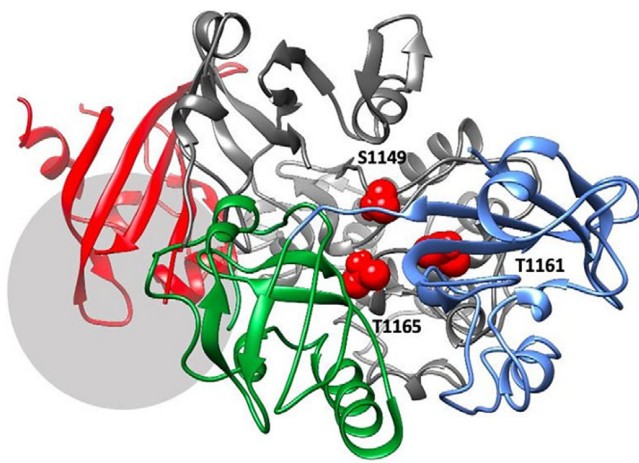

**Figure 4.** The best-ranking model of PAS domains interacting with the kinase domain of PASK. PASK domains are colored as follows: PAS1 – blue; PAS2 – dark green; PAS3 – red, kinase domain – dark gray. Residues S1149, T1161, and T1165, which are autophosphorylated in the activated kinase, are colored red and highlighted as spheres. The region downstream R1267, which contains several phosphorylated serine residues including S1280, S1287, and S1289, is marked by a light gray circle.

Both models share common structural features (Figure 3*a*, *b*), which can be used in further analysis of interactions between this domain and other domains of PASK. In addition, established models allow for the follow-up experimental work, which may include expressing different protein constructs and evaluating the function of those domains in cell models.

## Interactions between PAS and kinase domains

To explore possible interactions between all three PAS domains and the kinase domain within human PASK, we proceeded with the sequential protein–protein docking calculations, as described in

'Materials and methods' section. Because of several constraints, such as distances between PAS1–PAS2 and PAS2–PAS3 domains, this approach narrowed down and generated 19 potential models, as shown in supplementary information (Supplementary Figure S11 and Supplementary Table ST1). The models were cross-checked with the docked ternary complexes of three PAS domains (PAS1–PAS3) and poses that ranked highly in both calculations were prioritized for further analysis.

Even though the predicted energy scores are approximate and thus retain only indicative values, and models are highly speculative, the mechanism in which PAS domains are likely to modulate the PASK kinase domain has emerged. The results of protein–protein docking calculations indicate three binding 'hot spots' at the surface of the kinase domain, where the PAS domains are likely to bind (Figure 5). Those 'hot spots' were recurrent as highly ranked binding sites, retrieved by all docking runs. Structural constraints within PASK, such as very short PAS1–PAS2 linker and short PAS2–PAS3 loop, favored certain complexes over others.

Our results suggest that all three PAS domains may be involved in interactions with the kinase domain simultaneously (Figure 4). This model is supported by available mutagenesis data on PASK residues regulated by autophosphorylation: S1149, T1161, T1165, S1287, S1289, S1273, S1277, and S1280 (Kikani *et al.*, 2010). PAS1 interacts with T1161, PAS2 interacts with T1165 and S1149, while PAS3 interacts with residues downstream R1267. This is shown in Figure 4. Residues in PAS domains predicted to be important in maintaining interactions with the kinase domain are listed in Supplementary Table ST2. In future work, we plan to mutate those residues to further validate our model.

## Modulation of PASK by small-molecule ligands

A novel crystallographic screening method has been developed and reported recently by O'Rreilly *et al.* (2019). In this approach, the high-concentration aqueous soaks were made with a chemically

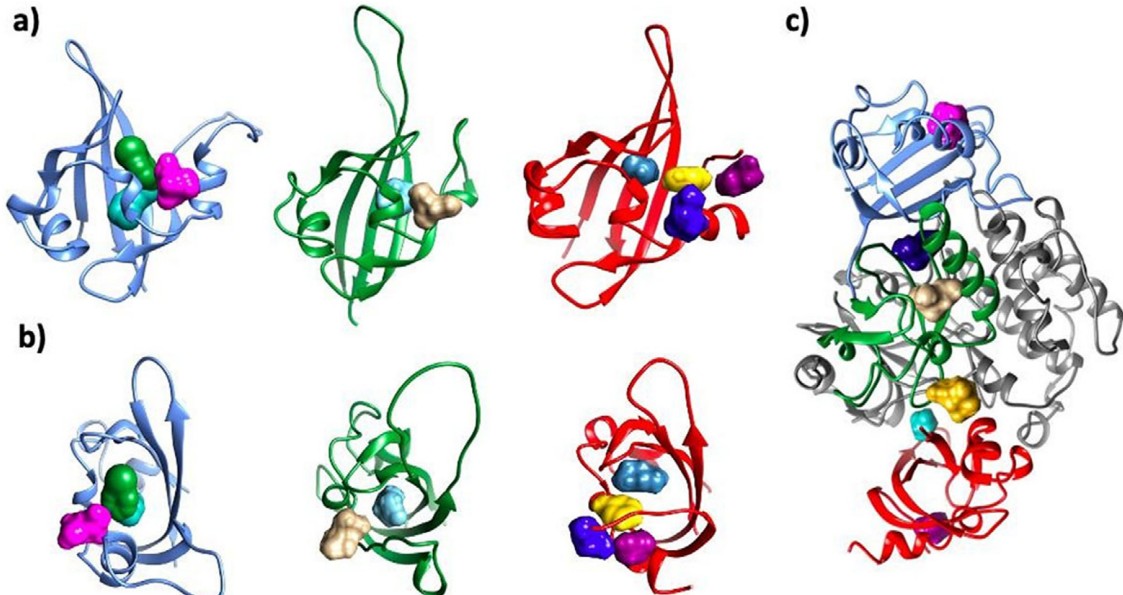

**Figure 5.** Consensus 'hot spots' identified on all three putative PAS domains and the interface between PAS and kinase domains. The following colors were used for each domain: PAS1 – blue; PAS2 – dark green; PAS3 – red; kinase domain – dim gray. For each PAS domain, two orientations are showed: side (*a*) and top (*b*). For the hotspots, the following colors were used: PAS1: H1 – bottle green; H2 – aqua; H3 – pink; PAS2: H4 – tan; H5 – cyan; PAS3: H6 – marine blue; H7 – yellow; H8 – electric blue; H9 – dark magenta. Kinase–PAS domains (*c*): H13 – navy blue; H14 – gold; H15 – turquoise. In panel *c*), location of probes H3 in PAS1 (pink); H4 in PAS2 (tan), and H9 in PAS3 (dark magenta) are shown for reference. Nomenclature of hotspots is consistent with Supplementary Table ST3.

diverse and ultra-low-molecular-weight fragment library denoted as 'MiniFrags' (heavy atom count 5–7). This allowed the identification of 'druggable' sites on proteins of interest. High screening hit rates reflected enhanced sampling of chemical space. MiniFrag screening can represent thus a highly effective method for 'druggability' assessment and guiding optimization of fragment-derived lead compounds. Such an approach is applicable for *in silico* screening. In this work, we tested it by employing fast ensemble docking of an enriched version of the MiniFrag library (AugMiniFrag; 94 compounds). The original MiniFrag library, which contains a set of chemically diverse fragments obtained from Enamine (www.enamine.net), was been augmented by fragments derived from amino acids known to modulate proteins (e.g., arginine), sugars, free fatty acids, and small drug-like molecules (see Section 'Materials and methods').

The set was used in molecular docking against all binding sites detected by FTMap and validated by the binding site mapping mode within SeeSAR (www.biosolveit.de). First, we used all individual domains as targets, using ensembles derived from PCA and cluster analysis of MD simulation trajectories as inputs. Next, we probed interfacial pockets, using outputs from MD simulations of the PAS1–PAS2 segment and results of protein–protein docking (PAS–kinase domains) as inputs.

Results are shown in Figure 5 and Supplementary Table ST3. All PAS domains under investigation were found to be 'druggable' by small molecules using FTMap and SeeSAR, including both alternative models of PAS3. The major difference between the two models of PAS3 is the nature of the binding site at the inner cavity of the PAS domain: while the binding site within PAS3 predicted by AlphaFold is very hydrophobic (all side chains but Y894, lining the central cavity, are hydrophobic), the model calculated by homology modeling has a more polar binding cavity (residues Y396, S399, N413, R420, Q428, D432, and K473; Supplementary Figure S12), able to bind a broad spectrum of chemically diverse small molecules, including phosphorylated sugars and side chains of free amino acids (Supplementary Table ST3). We suggest that future studies involving mutation of those residues may contribute to validation of the PASK model and further our understanding of its regulation by small molecules.

Inner pockets of PAS1 and PAS2 have a propensity to bind hydrophobic ligands; however, additional sites were detected on those domains, including the interfacial site at the protein–protein interaction (PPI) interface between both domains. Amezcua *et al.* (2002) reported nine compounds that bound PAS1 with dissociation constants smaller than 100 μM. Those compounds were hydrophobic in character, containing one or two aromatic rings substituted with polar groups around their peripheries, and separated by a short linker (0–2 heavy atoms). Molecular docking of those six compounds into the central cavity of PAS1 reproduced those measured *Kd* values well (p-value = 0.0029; Supplementary Figure S16 and Supplementary Table ST4). Furthermore, molecular docking of the enriched MiniFrag set identified two distinct binding sites on PAS1, additional to that central cavity: one of those sites was modulated by simple sugars (Supplementary Table ST3). Collectively, we identified >10 consensus 'druggable hotspots' involving all three PAS domains of PASK. Interfacial binding sites were found on each PAS domain of PASK, which may be explored in the development of allosteric inhibitors of PASK in the future.

We inspected the predicted interface between the kinase domain and PAS domains. Three 'hot spots' were detected at the interface between PAS and kinase domains, and the site comprised of three domains: PAS1, PAS2, and kinase domain (Figure 5). These two

sites were found to bind arginine and metformin. Metformin is a first-line therapy for the treatment of diabetes type 2 due to its robust glucose-lowering effects and well-established safety and tolerability (Diabetes Prevention Program Research Group, 2012). However, its mechanism of action is poorly understood. Its putative mechanisms of action include indirect activation of AMPK, inhibition of cAMP production, modulation of mTOR, blocking the action of glucagon, and activation of TAK1-IKKα/β signaling (Agius *et al.,* 2020). However, the main protein target of metformin remains unknown, and metformin's activity may rely on modulating multiple targets. Since PASK is regulated by the activity of AMPK and mTOR (Hurtado-Carneiro *et al.,* 2014), it is plausible that it can be among the targets modulated by metformin, and its direct interactions should be followed up experimentally. If it were determined that PASK is indeed directly modulated by metformin, it would be beneficial to evaluate whether metformin would indeed the kinase as well as PAS domains of PASK, and whether such binding would disrupt the PPI interface, or make the interactions between the domains stronger.

## Conclusions

In this work, we present results of an integrative computational study on human PASK kinase, combining structural bioinformatics, molecular modeling, all-atom MD simulations, molecular docking and 'hot-spot' mapping. Our results strongly suggest the presence of three consensus N-terminal PAS domains in human PASK kinase. The experimental structure of one of those domains has resolved by solution NMR and two other domains are predicted; future work will focus on experimental confirmation of those domains. All domains are predicted to be 'druggable' by small molecules, albeit with different domains likely to be modulated by different classes of ligands. This is likely to contribute to PASK's role as a versatile metabolic sensor. Predicted binding sites for diverse small molecules include sites within individual PAS domains as well as 'druggable hot spots' at domain–domain interaction interfaces. Proposed three-dimensional model of interactions between PAS domains and the kinase domain of PASK fits with experimental data available and consolidates the model of the kinase domain being inhibited by direct interaction with PAS domain; these interactions may be modulated by small molecules. Future work will focus on those small molecules and their binding sites, which may pave the way for future drug discovery in not only liver and metabolic disorders but also other diseases influenced by oxidative stress, such as neurodegenerative disease. Increased understanding of PASK and its modulation by small molecules has the potential to profoundly impact healthcare research.

**Open peer review.** To view the open peer review materials for this article, please visit http://doi.org/10.1017/qrd.2024.1.

**Supplementary material.** The supplementary material for this article can be found at http://doi.org/10.1017/qrd.2024.1.

**Acknowledgments.** We are grateful to Dr S. Kometa (HPC Rocket) and Dr C. Wills (Newcastle University NMR Facility) for their technical assistance.

**Author contribution.** P.Z. and A.K.B. outlined the problem and conceptualized the study; J.V.d.S., K.S.H., and A.K.B. designed the study; L.F., A.K.B., and J.V.d.S. performed the bioinformatic analysis; R.Z.; H.J., and L.F. carried out the equilibrium atomistic molecular dynamics simulations of PAS1 and PAS2 domains; S.X. and P.Z. carried out the equilibrium atomistic molecular dynamics simulations of PAS3 and PD4 domains; J.V.d.S. and S.X. wrote scripts for data analysis; S.X., L.F., P.Z., R.Z., and H.J. extracted and analyzed data from the

simulations; R.Z., H.J., A.K.B., and K.S.H. performed molecular docking calculations; A.K.B. and K.S.H. supervised the study; all authors produced the figures, plots, and supplementary data. J.V.d.S. and A.K.B. wrote the first draft of the manuscript. K.S.H. edited the manuscript. All authors have read and agreed to the published version of the manuscript.

**Financial support.** This work was funded by a EPSRC Ph.D. studentship to L.F. (EPR51209X1) and EPSRC funding to A.K.B. (EP/S022791/1).

**Competing interest.** The authors declare no competing interests.

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
