## [Reviewer Report]

QRD-D-22-00006

The paper describes the application of a sophisticated collection of structural bioinformatic tools and biomolecular simulation methods to the prediction of the structure and ligand-binding characteristics of the human PASK kinase. Though not in itself proving a definitive answer to this question, it provides a rich set of testable hypotheses for further theoretical and experimental studies, via which this might be achieved.

The work seems to have been done to a high standard, the methodologies used are well described and the results are discussed critically. There are just a few areas where I think some improvements are necessary, or would be very worthwhile:

1. For the homology modelling of individual domains, the sequence alignments with the chosen templates and other metrics (e.g. % sequence identity) should be added to the supplementary material.

2. For the protein-protein docking studies, it would be good to show, if possible, that the chosen methodologies have reasonable performance when applied to related systems where the correct answer is known. For example, they could be tested on other examples of kinases with separate regulatory subunits (maybe protein kinase A, but there could be better examples).

3. For the protein-ligand docking studies, the validation of the methodology through the correlation with the results of Amezcua et al. is claimed but not shown - it should be. In addition, it would be good if it can be shown that related molecules that were found by Amezcua et al. to be much weaker binders are also predicted correctly (increasing the confidence that the approach avoids false positives).

4. For the molecular simulations, using RMSD to evidence equilibration is not a reliable approach. Even when an RMSD from an initial conformation has plateaued, the molecule can be undergoing continued and significant conformational change. A much better approach is to use cluster analysis to produce a discretized version of the trajectory, and plot this as a time series. Cluster analysis is mentioned at a few points in the manuscript, but I don’t see any examples of it in the results. A related point is that details of the clustering method should be included.

5. At some point I would like to see a more explicit comparison of the subunit organisation predicted in this study with that predicted by Alphafold. Though Alphafold-derived structures are mentioned at various points this seems to be with regard to structure prediction of individual subunits rather than their relative positioning in the complex as a whole. The Alphafold model has a large amount of low-confidence structure, but does it include anything that might be pointing to where PAS2 and PAS3, in particular, sit?

---

## [Reviewer Report]

In this study the authors claim that the molecular mechanism of PASK' activity and the structural information are not well understood. Therefore, they investigate these gaps with molecular modelling and simulation tools. In terms of structural information, the authors were interested to know the number of PAS domains PASK is likely to have. In terms PASK’s activity the authors were interested to know how the discovered domains modulated the kinase activity and how the domains could be potentially controlled by a small molecule.

The authors reported there were three N-terminal PAS domains. By solvent mapping and fragment docking the authors found potential druggable interfaces within the domains. The authors also suggest mention that they were able to identify mechanism of auto-inhibition of kinase activity, that correlated to the presence of all three PAS domains that they identified. Further addressing that in future work they will improve the “model” and develop small-molecule inhibitors of PASK by targeting the druggable hot-spots.

This reviewer finds the work exciting but however has few concerns and confusion regarding the author’s manuscript as listed below.

1. In the abstract, could the authors clarify what they mean by “Future work will focus on validation of this model ...”. The reviewer is primarily confused about model the authors are referring to.

2. In the introduction, could the authors provide a citation for the sentence “Despite its evolutionary conservation... the molecular mechanism underlying PASK regulation remains poorly understood ...”

3. While the authors claim that PASK’s structural information is not well understood (in abstract), later (in introduction) the authors' report “crystal structure of the kinase domain of human PASK shows that this domain adopts an active conformation and has catalytic activity in the absence of activation loop phosphorylation (Kikani et al., 2010).”. These two sentences are in contrast to each other. Therefore, the reviewer requests the authors to revise their abstract.

4. This reviewer also finds the term “AI modelling” in the sentence “Three-dimensional models of those domains were calculated using a combination of homology modelling, threading, and AI modelling by AlphaFold (Jumper et al., 2021)” misleading. The authors simply used a deep learning tool to predict the structure. Furthermore, the authors only cite what they call AI modelling, but does not mention or cite what tools they use for homology modelling and threading. Therefore, the reviewer is requesting for a revision in this sentence.

5. Could the author provide a citation to the mentioned experimental data in the sentence “Such a model is consistent with the experimental data available” and/or elaborate on the type of experimental data and so on.

6. While the authors have explored the interaction of PAS and kinase domains by protein-protein docking, the recent release of AlphaFold-Multimer has shown that it outperforms protein-protein docking by a large margin. Could the author repeat the experiment with AlphaFold-Multimer and carry out similar analysis and also report how they are similar or differ from their results from protein-protein docking?

7. Could the authors cite this “... a first-line drug to treat type 2 diabetes, ...”

8. What parameters were used for AlphaFold? If this the default settings, the authors should mention this in their methods section. Additionally in the introduction the authors mention threading separately however in the methods section no details regarding this was found? Could the authors comment more on why this is the case? Was threading part of the homology modelling procedure of Phyre2? If so then the authors should clarify this. If not, then the authors need to provide additional details.

9. In the structural bioinformatics method section, the author several methods and provide the link to their website. However, the link “http://www.tigr.org/tdb/tgi” report page not found. The authors should double check if the link they provided is the correct link. The reviewer also believe citing the methods paper for these tool maybe enough for readers. Furthermore, could the authors expand this section and explain exactly what tools were used for what as well as the parameters that were set when these tools were used. Was this default or customized? If default, then this needs to be mentioned. If customized then it needs to mentional as well as rationalized. Additionally a schematic of a workflow diagram for this section would make it very clear for readers as to how the authors used these tool. The reviewer strongly suggest in making such schematic.

10. For the sentence “... using the following templates: 1LL8 (PAS1); 5SY5 (PAS2); 4EH0 (PAS3) ...” could the author mention what they mean by PAS1, PAS2, and PAS3? The reviewere is assuming that these are the three domains? The authors need to clarify this. Furthermore, in the sentence mentioned, the authors also need to mention that the code outside the parathesis are PDBID (assuming they are that), as it may not be clear to readers who may not know what these are.

11. In the data analysis section, the authors need to correctly cite FTMap as mentioned in their publication page of their website “http://ftmap.bu.edu/publications.php”.

12. The authors need to mention what type of clustering algorithm they used. If this is gmx cluster then they need to cite the clustering algorithm with “Angew. Chem. Int. Ed. 1999, 38, pp 236-240”.

13. As mentioned before, in the protein-protein docking section, the reviewer is curious to why the authors chose not to use AlphaFold-Multimer. In this way sampling billions of conformations could have been avoided. The reviewer urges the authors to carryout complex structure prediction with AlphaFold-Multimer, carryout all analysis and report on their findings.

14. Could the authors add a sentence as for the line “Flexible loops and annotated ntrinsically disordered regions were excluded from the calculations” as to why this was done?

15. The reviewer suggests all complex structure prediction to be carried out with AlphaFold-Multimer.

16. While the authors mention the usage of Swiss Model in Figure 3b. They do not mention the protocols they used for Swiss Model in their Methods sections. Therefore the reviewer request the authors to revise their Methods section and include the procedures as appropriate.

17. Could the authors provide additional details as to why PD4 partially unfolded?

18. In the Results and Discussion section, the authors mention “H-bond analysis of both models ...”. Could authors elaborate how the H-bond were analyzed. What was used to analyze the H-bond in these trajectories. Details in the methods section and briefly remind readers of what was used in the results and discussion section.

19. Could the authors add more discussion sentences as to why this was the case for the line “AlphaFold model was considerably more stabilised by intramolecular H-bonds than the homology model”?

20. For the sentences “The analysis of potential energies for both models favoured the AlphaFold model: -5000 ± 300 kJ/mol on average, compared with -3000 ± 400 kJ/mol for the homology model. These results strongly indicate that the existence of a third PAS domain within PASK is very likely.”, could the authors clarify further why this indicated the existence of a third PAS domain? Additionally could the authors remind the readers how the potential energy calculation were done?

21. For Fig. S10, could the authors add a few discussion sentence as to why there was high fluctuation for the AlphaFold model around the time 50ns - 65ns but not for the homology model. This is counter intuitive given the report that there were higher number of H-bonds for the AlphaFold model than the homology model. Could author report why this was the case?

22. Could the authors align the AlphaFold model and the homology model and report the structural similarity (RMSD) in Fig. 3a -3b.

23. In Fig. 4, the author talk about experimental data, but it is not clear what these experimental data are. Could the authors please clarify what experimental data they are discussing?

24. While the author used deep learning tool (AlphaFold) to predict the structure of PAS domain, they did not use any such deep learning tool to identify druggable hotspots. The reviewer urges the authors to make use of some of the well known open-source deep learning tools to further validate druggable hotspots within the PAS domains.

25. In the following line “Molecular docking of those nine compounds into the central cavity of PAS1 reproduced those measured Kd values well (data not shown)”, the reviewer is intrigued as to why the correlation of the Kd values were not show and urges the authors to provide additional comments regarding this. Additionally how were the Kd values predicted? The authors need to clarify this.

Overall, this reviewer finds this work very exciting and is looking forward to the changes/revisions requested.